# Grape Stalk Valorization: An Efficient Re-Use of Lignocellulosic Biomass through Hydrolysis and Fermentation to Produce Lactic Acid from *Lactobacillus rhamnosus* IMC501

Sergio D'ambrosio [1], Lucio Zaccariello [2], Saba Sadiq [2], Marcella D'Albore [2], Giovanna Battipaglia [2], Maria D'Agostino [1], Daniele Battaglia [2], Chiara Schiraldi [1] and Donatella Cimini [2,*]

1   Section of Biotechnology, Medical Histology and Molecular Biology, Department of Experimental Medicine, University of Campania L. Vanvitelli, Via de Crecchio 7, 80138 Naples, Italy; sergio.dambrosio@unicampania.it (S.D.); chiara.schiraldi@unicampania.it (C.S.)

2   Department of Environmental, Biological and Pharmaceutical Sciences and Technologies, University of Campania L. Vanvitelli, Via Vivaldi 43, 81100 Caserta, Italy; lucio.zaccariello@unicampania.it (L.Z.); giovanna.battipaglia@unicampania.it (G.B.)

*   Correspondence: donatella.cimini@unicampania.com; Tel.: +39-081-5664345

**Abstract:** *Lactobacillus rhamnosus* is a homofermentative probiotic strain that was previously demonstrated to grow on lignocellulosic-derived raw materials and to convert glucose into L-lactic acid (LA) with yields that vary between 0.38 and 0.97 g/g. Lactic acid is a key platform chemical, largely applied in different biotechnological fields (spanning from the pharmaceutical to the food sector) and also as a building block for the production of biodegradable polymers. In the present study, grape stalks were evaluated as sources of fermentable sugars for the growth of *L. rhamnosus* IMC501 and for the production of LA, since millions of hectoliters of wine are produced every year worldwide, generating a huge amount of waste. Although grape stalks are quite recalcitrant, the combination of a steam explosion pre-treatment with optimized two-step hydrolysis and commercial enzymes (Cellic-CTec2) allowed us to obtain a cellulose conversion efficiency of about 37% and to develop small-scale 2 L batch fermentation processes. Results successfully demonstrate that *L. rhamnosus* IMC501 can tolerate biomass-derived inhibitors and grow on grape stalk hydrolysate without the need for additional sources of nitrogen or other nutritional elements, and that the strain can convert all glucose present in the medium into LA, reaching the maximal theoretical yield.

**Keywords:** lactic acid; *Lactobacillus rhamnosus*; grape stalks; circular economy; steam explosion; biorefinery; sustainability

## 1. Introduction

Lignocellulosic biomasses (LBs) represent a type of feedstock recovered from different resources such as natural forests, agricultural production fields, and agricultural wastes [1]. This widely available raw material contains compounds such as polysaccharides (cellulose and hemicellulose), which can be an energy source and/or can be converted into value-added products, such as biochemicals, from a circular economy "zero-waste" perspective [2]. To obtain fermentable sugars, the cellulose and hemicellulose need to be exposed and hydrolyzed with acids or enzymatic treatments or by using cellulase- and hemicellulase-producing microorganisms [3]. To modify the refractory structure of lignocellulose and improve biomass digestibility, various pre-treatment strategies that include chemical, physical, physicochemical, and biological approaches have been developed [4]. Based on the source and composition of the biomass, results can largely vary, and the severity/harshness of the treatment also affects the composition of the hydrolysate in terms of sugars and inhibitors potentially released [4]. Lignocellulosic feedstocks were used for numerous biotechnological purposes in different fields [5], with a focus on the production

of bioactive compounds and the identification of alternative processes for biofuel production [6]. The huge quantity of petrochemical plastics used and present in grounds and seas recently also prompted the rapid identification of biotechnological strategies that reduce environmental pollution, including the production of bioplastic from renewable waste resources [7,8].

Among waste products, grape stalks (5–7%) are recovered after destemming grapevines during winemaking, and therefore their output is directly associated with the manufacturing process [9]. This makes them a key non-food winery by-product with a high content of lignin and holocellulose that can be extracted and used as sources of antioxidant compounds for biomedical applications [10] and as substrates for fermentation processes [11], respectively. Based on the information collected in twenty-nine countries that represent 91% of the global market, in 2022, global wine production was estimated between 257.5 and $262.3 \times 10^6$ hL, with a mid-range estimate of $259.9 \times 10^6$ hL. For this reason, the re-use of grape stalks in circular economy approaches has been investigated so far [11]. Europe, in particular, is estimated to produce $157 \times 10^6$ hL of wine, 32% of which is produced in Italy (OIV 2022) [12].

Among organic compounds that can be obtained through fermentation processes, lactic acid (LA) is a key platform chemical with established uses in the food, cosmetic, pharmaceutical, and chemical industries [13,14]. Recent applications as a building block in the production of biodegradable polymers for the large-scale replacement of petrochemical plastics [15] further underlined its role and kept attracting industrial interest. The common industrial production of lactic acid is based on microbial fermentation (from lactic acid bacteria) because it is chemically and economically more feasible compared with chemical production approaches and enables the production of an optically pure lactic acid [16]. Many attempts to produce LA by replacing pure glucose with cheap carbon sources such as orange peels, molasses, and wood processing waste (after pre-treatment and chemical or enzymatic hydrolysis) are reported in the literature [13,17].

Grape stalks were previously evaluated as a fermentation substrate to produce LA from *Rhyzopus oryzae* growing on solid medium (agar and grape stalks) [18]. The advantages of this fungus are the low nutritional requirements and the high LA yield, which is, however, strongly affected by the high broth viscosity and resistance to oxygen transfer [19].

*Lactobacillus rhamnosus* strains can use glucose for homolactic fermentation, producing pure L-LA, with yields on different lignocellulosic raw materials ranging from 0.38 up to 0.97 g/g [13].

In the present study, the production of lactic acid from the probiotic strain *L. rhamnosus* IMC 501 [20–23] on a non-agricultural biomass massively present in the Campania region (grape stalks) was tested to evaluate its suitability as a raw material from the perspective of a biorefinery concept, supporting environmental and economic sustainability of biotechnological processes.

## 2. Materials and Methods

### 2.1. Biomasses and Medium

Grape stalks were provided by a wine-distillery facility located in the Campania region of Italy. The probiotic strain *L. rhamnosus* IMC501 was supplied by the "Centro Sperimentale del Latte S.r.L." (Zelo Buon Persico, Italy). All medium components and salts, as well as ammonium hydroxide, were supplied by Sigma-Aldrich (St. Louis, MO, USA). Yeast extract was furnished by Organotechnie (La Corneuve, France), while sulfuric acid was purchased by Biochem s.r.l. (Turin, Italy). The semi-defined medium (SDM) used for control growth experiments contained per liter: 10 g of yeast extract, 10 g of soy peptone, 0.25 g of $MgSO_4*7H_2O$, 0.05 g of $MnSO_4*H_2O$, 2 g of $Na_3C_6H_5O_7$, 0.45 mL of Tween 80, 0.5 g of L-ascorbic acid, and 0.2 g of NaCl. Glucose (Sigma-Aldrich, St. Louis, MO, USA), used as a carbon source, was autoclave sterilized and added to the semi-defined medium.

### 2.2. Quantification of Cellulose and Hemicellulose in Grape Stalks

Hemicellulose and α-cellulose were extracted from 5 samples using the method introduced by Loader and colleagues [24] and modified according to [25]. The procedure is based on a triple-step digestion: in the first step, resin, fatty acids, and ethereal oils were extracted with a solution of ethanol/toluene (50:50 *v/v*) for 7 h; later, lignin was removed using iterative washes of an acidified sodium chlorite (7% *w/v* NaClO$_2$) solution for a minimum of 36 h to obtain holocellulose. Third, the soluble portion of holocellulose (hemicelluloses and β-cellulose) was removed with sodium hydroxide (5% *w/v* NaOH solution for 2 h at 60 °C, twice) to leave the insoluble α-cellulose.

### 2.3. Biomass Pre-Treatment and Hydrolysis

Grape stalks (GSs) were pre-treated using three different protocols:

(1) Alkaline hydrolysis: briefly, after size reduction in a benchtop grinder, dry biomass was dissolved in ammonium hydroxide (10% *v/v*) at 10% (*w/v*) solid loading and incubated at 70 °C for 22 h. The pH was adjusted to 7 with hydrochloric acid before biomass recovery by centrifugation and washing with water (twice). The supernatant was then discarded, and the pellet was placed in an oven at 70 °C for 24 h [26].

(2) Steam explosion (SE): the SE pre-treatment of grape stalks was conducted in a 1.5 L stirred-batch vessel made of AISI 316L. The vessel is heated by three electric heating elements of 0.7 kW each. The temperature is guaranteed by a control loop using a type K thermocouple connected to the internal wall of the vessel, a comparator for the set-point temperature, and a voltage controller for tuning the current into the resistance. A 3-cm-thick insulating layer of glass wool is used to minimize vessel heat dissipation. In the SE process, the biomass was heated with water for several minutes under autogenous pressure and then subjected to a sudden pressure drop. This led to vapor expansion inside the biomass, determining the disruption of the biomass fibers' structure and thus making cellulose more accessible. For each SE experiment, 1200 mL of deionized water and 200 g of dried biomass were used. Parameters recorded during the steam explosion experiments are displayed in Figure 1, which shows that the water/biomass mixture was heated from 40 °C to 165 °C with a heating rate of about 2 °C/min. At the set temperature, a pressure of 8 bars was recorded. Once the temperature set-point was reached, the ball valve at the bottom of the vessel was opened, and the biomass was exploded/discharged into a collecting tank.

(3) A sequential combination of both. Ammonium hydroxide pre-treatment on steam-exploded grape stalks.

All hydrolysis experiments on the pre-treated biomasses were performed at a temperature of 50 °C and lasted 24–48 h. Different amounts of Cellic CTec2 (donated by Novozymes, Bagsvaerd, Denmark) were evaluated to identify the best hydrolytic conditions. In a first set of tests, 24 h hydrolysis experiments were carried out on 10% (*w/v*) pre-treated biomasses suspended in 20 mL of sodium acetate buffer (5 mM) at pH 5.2 with 2.7% *w/w* (g$_{enz}$/g$_{cellulose}$). These experimental conditions were used on grape stalks treated by (i) SE, (ii) alkaline hydrolysis, and (iii) the combination of SE and alkaline hydrolysis to select the most convenient approach. Successive hydrolysis tests were performed on grape stalks only pre-treated by steam explosion. In particular, experiments aimed to identify the optimal amount of biomass and enzyme with respect to cellulose content and the timing of enzyme addition (one pulse, two pulses). In the second set of tests, 2.7% *w/w* (g$_{enz}$/g$_{cellulose}$) (120 μLl) and 5.5% *w/w* (g$_{enz}$/g$_{cellulose}$) (250 μL) were added either only at time zero of the experiment or also after 24 h of incubation. In both cases, the biomass load was equal to 10% (*w/v*). In the last set of experiments, the hydrolysis was carried out on a higher concentration of pre-treated biomass equal to 15% (*w/v*) with 250 μL of enzyme added twice (at time zero and after 24 h), thereby reducing the ratio of % g$_{enz}$/g$_{cellulose}$ to 3.6 + 3.6.

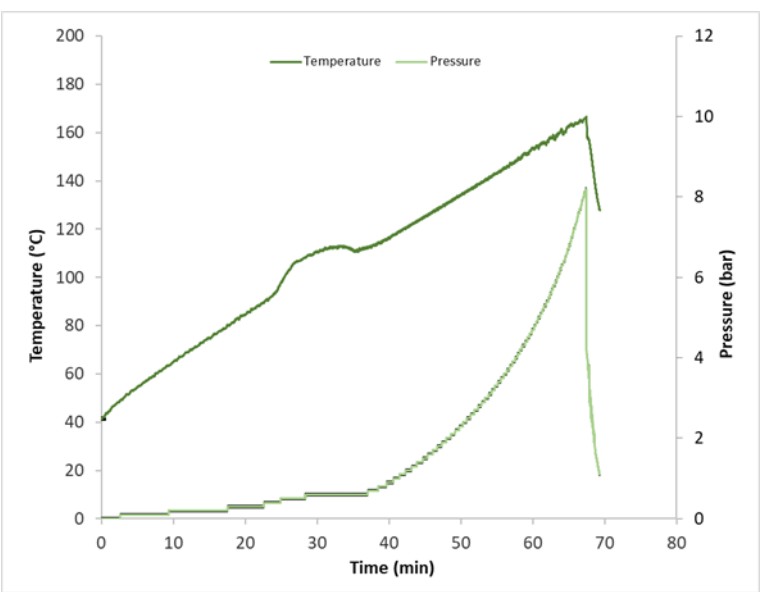

**Figure 1.** Temperature and pressure profiles during SE experiments.

*2.4. Morphological Analysis of Grape Stalks by Scanning Electron Microscopy (SEM)*

The morphology of samples pretreated by steam explosion and alkaline hydrolysis and the morphology of the untreated control were evaluated by SEM. The samples were fixed in 4% (*w/v*) paraformaldehyde and dehydrated with increasing ethanol concentrations (30, 50, 70, 90, 95, and 100% *v/v*). The air-dried samples were mounted on aluminum stubs, sputter-coated, and vacuumed with a platinum-palladium Denton Vacuum (DESK V). FESEM (Field-Emission SEM) Supra 40 (ZEISS, Jena, Germany; EHT = 5.00 kV, WD = 22 mm, detector in lens) was used for observation.

*2.5. Small-Scale Bottle Experiments*

A 20% (*v/v*) glycerol stock solution of *L. rhamnosus* IMC501 was inoculated into 0.1 L of SDM medium in a 0.1 L pyrex screw-cap bottle to reach a starting concentration of 0.1 $OD_{600}$. The bottle was incubated in a rotary air shaker incubator (model Minitron, Infors, Bottmingen, Switzerland) at 37 °C and 150 rpm. Small-scale experiments lasted 48 h. SDM medium (Section 2.1) was used as a control to compare the growth of *L. rhamnosus* on grape stalk hydrolysates with and without supplementation of 0.1% (*w/v*) of yeast extract. Samples were withdrawn from the culture to analyze glucose consumption and acid production. All experiments were conducted at least in triplicate.

*2.6. Batch Experiments in Controlled Bioreactor Conditions*

Batch experiments were run in duplicate on a Biostat CT plus reactor (3.2 L total volume) with a working volume of 2.3 L (Sartorius Stedim; Gottingen, Germany) provided with a digital control unit and connected to a PC for remote control using MFCS-win software. A stock of *L. rhamnosus* ICM501 was inoculated in 0.25-L bottles on SDM medium with glucose as a carbon source. Once the strain reached the exponential phase, the whole preculture was transferred to a Biostat CT plus reactor containing steam-exploded grape stalk hydrolysate at 10% (*w/v*) with and without supplementation of complex nitrogen sources, namely 0.1% (*w/v*) of yeast extract and 1× Salts, in order to have a starting glucose concentration of 19 g/L. Batch experiments lasted 24 h, the temperature was controlled at 37 °C, and a constant pH equal to 6 was maintained by adding 5 M NaOH and 30% (*v/v*) $H_2SO_4$. The stirring velocity was 150 rpm, and an air supply of 0.44 vvm was applied during the course of the fermentation (percentage of dissolved oxygen recorded by the electrode during the process: decreasing from 100 to 0% for the first three h and stable at 0% for the remaining 21 h of growth). For the duration of all fermentations, 10–20 mL

of broth samples were withdrawn from the bioreactors every 2 h to determine substrate consumption and lactic acid production. Samples collected were centrifuged (model ALC PK 131R, Labexchange, Burladingen, Germany) to separate the biomass from the supernatant at 6500 rpm for 10 min at 4 °C. The supernatant obtained after centrifugation was ultrafiltered on 3 kDa centrifugal filter devices (Centricon, Amicon, Sigma-Aldrich) at 12,000 rpm for 10 min at 4 °C in an Eppendorf Microcentrifuge 5415R (Hamburg, Germany). Permeates were analyzed to determine and quantify sugars, organic acids, and phenolic compounds by High-Pressure Liquid Chromatography (HPLC).

### 2.7. Quantification of Glucose, Organic Acids, and Potentially Inhibitory Compounds by HPLC

Permeates were analyzed to determine glucose, xylose, and lactic acid produced during growth by HPLC (UHPLC Dionex Ultimate 3000; Thermofisher, Waltham, MA, USA) on an LC Column Phenomenex Rezex$^{TM}$ ROA-organic Acid H+ 8% (300 mm × 7.8 mm), 6 μ at 40 °C. Analyses were performed by isocratic elution with 0.1% ($v/v$) $H_2SO_4$ in ultra-pure water as the mobile phase at a flow rate of 0.8 mL/min and an acquisition time of 25 min. Detection was performed using UV absorbance at 200 nm, a data collection rate of 2.0 Hz, a time constant of 0.60 s, and a refraction index (Shodex RI-101 detector: Step: Auto; Max. Auto Step: 5.1 s; Average: On; Temp. Nominal: 32 °C; Rise Time: 1.0 s; Polarity: Plus; Record Range: 512.00 μRIU; Integrator Range: 500 μRIU/UV). Peak areas were evaluated through the Thermofisher Chromeleon Software. Version 6.80 A mix of standards containing glucose, xylose, acetic acid, lactic acid, and succinic acid was obtained from Supelco (Merck KgaA, Darmstadt, Germany) for the quantification of organic acids and sugars. Standard solutions in the range of 30.0–0.23 g/L were used to test the linearity of the analytical method, sensitivity, and reproducibility. The standard solutions were injected (10 μL), and the areas were plotted versus the amount of injected sample to obtain the calibration curves. The steam-exploded and hydrolyzed grape stalk slurry used for bottle and fermenter experiments was analyzed for the determination of furfural, vanillin, and 4-hydroxybenzoic acid according to the previously described method [27]. The concentration of inhibitory compounds was also analyzed at the end of the experiment, after 24–48 h of growth.

## 3. Results

### 3.1. Pre-Treatment and Hydrolysis of Grape Stalks

Triple-step digestions of grape stalks (paragraph 2.2) indicated the presence of 33 ± 1.5% and 27 ± 2% of cellulose and hemicellulose, respectively. After size reduction, grape stalks were pre-treated by steam explosion and alkaline hydrolysis and evaluated as a potential source of fermentable sugars. Morphological analyses highlight a loss of compactness of the fibers after the pre-treatments as compared to the untreated sample (Figure 2).

To identify the most efficient pre-treatment conditions for GS, 24-h hydrolysis trials were carried out on the biomass pre-treated with three different methods to compare cellulose exposure for hydrolysis. The experiments were performed by adding 2.7% $w/w$ ($g_{enz}/g_{cellulose}$) of the commercial enzymatic mix to 10% ($w/v$) pre-treated biomass.

The results reported in Figure 3 show that similar concentrations of glucose were obtained from alkaline hydrolysis and steam explosion (t-student, $p_{value} > 0.05$) and that the sequential use of both procedures did not improve the results achieved. The following hydrolysis trials were performed by modifying the biomass and enzyme load on steam-exploded grape stalks. Conditions are reported in Table 1.

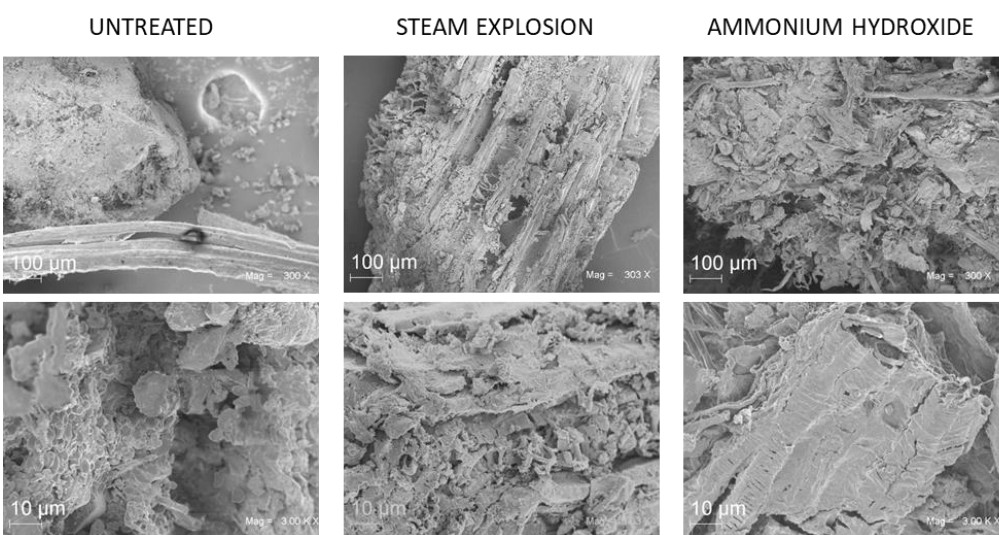

**Figure 2.** Morphological analyses of untreated and pre-treated grape stalks by scanning electron microscopy.

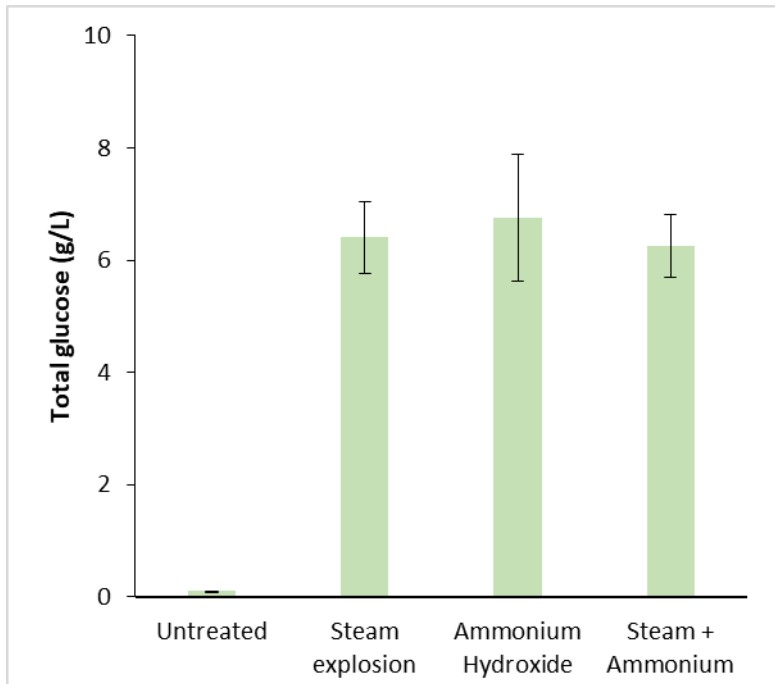

**Figure 3.** Enzymatic hydrolysis of differently pre-treated grape stalks.

A first set of experiments was carried out on 10% ($w/v$) pre-treated biomasses (t1–4). Then, 48-h hydrolysis experiments were conducted by supplementing the enzyme (2.7 or 5.5% $w/w$ $g_{enz}/g_{cellulose}$) either once at time zero of the experiments or by repeating the addition after 24 h with the same enzyme concentration. After 48 h, the concentration of sugars released was determined by subtracting the initial amount of glucose present in the enzyme (Figure 4). Two-step hydrolysis increased by 45 (t2) and 66% (t4), respectively, the concentration of glucose released after 48 h in comparison to what was present after the first 24 h of incubation.

**Table 1.** Conditions used for grape stalk hydrolysis experiments and the resulting yield of cellulose bioconversion.

| Test | Conditions | Bioconversion Yield (%) |
|:---:|:---|:---:|
| t1 | Biomass 10% ($w/v$) Enzyme 2.7% $w/w$ ($g_{enz}/g_{cellulose}$) | 18.7 |
| t2 | Biomass 10% ($w/v$) Enzyme 2.7 + 2.7% $w/w$ ($g_{enz}/g_{cellulose}$) | 27.8 |
| t3 | Biomass 10% ($w/v$) Enzyme 5.5% $w/w$ ($g_{enz}/g_{cellulose}$) | 32.8 |
| t4 | Biomass 10% ($w/v$) Enzyme 5.5 + 5.5% $w/w$ ($g_{enz}/g_{cellulose}$) | 38.1 |
| t5 | Biomass 15% ($w/v$) Enzyme 3.6% $w/w$ ($g_{enz}/g_{cellulose}$) | 16.4 |
| t6 | Biomass 15% ($w/v$) Enzyme 3.6 + 3.6% $w/w$ ($g_{enz}/g_{cellulose}$) | 22.9 |

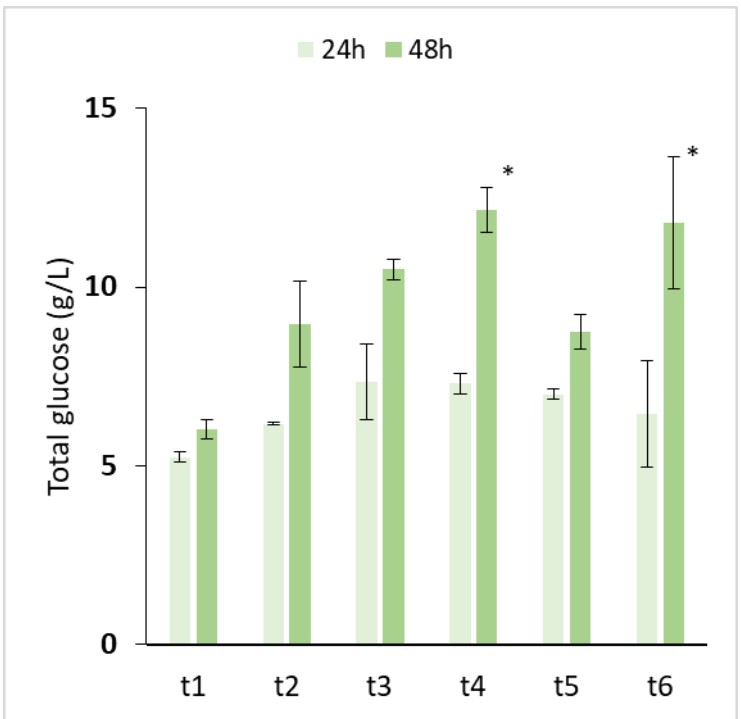

**Figure 4.** Hydrolysis of steam-exploded grape stalks after 24 and 48 h of incubation with Cellic Ctec2. Concentration of released glucose after 24 and 48 h of incubation. Comparison of results obtained with different biomass loadings (10 and 15% $w/v$), and enzyme concentrations (2.7, 3.6, and 5.5% $w/w$ $g_{enz}/g_{cellulose}$). * indicates a $p < 0.05$ between 24 and 48 h of incubation calculated by a two-tailed paired t-student test.

Similar results in terms of cellulose conversion efficiency and final glucose concentration were obtained when a total of 5.5% $w/w$ ($g_{enz}/g_{cellulose}$) was added either as a single pulse at time zero of the experiment (t3) or split into two separate additions (2.7 + 2.7% $w/w$ $g_{enz}/g_{cellulose}$, t2). The highest cellulose conversion (37%) was obtained by the repeated addition of 5.5% $w/w$ ($g_{enz}/g_{cellulose}$).

To further improve this result, the same condition was tested on a higher concentration of biomass (Figure 4, t5 and t6). Therefore, this second set of hydrolysis tests was carried out on 15% ($w/v$) steam-exploded grape stalks. The volume of enzyme employed in one-step (t5) and two-step (t6) hydrolysis experiments was the same as that used in previous trials

(t3 and t4), thereby leading to lower ratios of enzyme per gram of cellulose, namely 3.6 and 7.2% (3.6 + 3.6) $w/w$ ($g_{enz}/g_{cellulose}$). Unexpectedly, no improvement was observed.

### 3.2. Growth of L. rhamnosus IMC501 on Grape Stalk Hydrolysate in Bottle and Bioreactor Batch Experiments

Hydrolysates obtained under conditions set in t4 tests were used as substrates for the growth of *L. rhamnosus* IMC501 in small-scale bottle experiments. The growth of the probiotic strain was evaluated on the hydrolysate with and without a supplementation of 0.1% ($w/v$) of yeast extract. The semi-defined SDM medium, containing YE and soy peptone as complex nitrogen sources, was used as a control. Due to the presence of solid residues in the hydrolysate, growth on the latter was only evaluated by measuring the concentration of residual glucose and that of the produced lactic acid.

Data show that the concentration of lactic acid obtained on the hydrolysate-based medium is identical to that obtained on the control SDM medium (Figure 5), although the yield ($Y_{LA/glu}$) of about $1.09 \pm 0.02$ g/g on the control medium decreased to $0.86 \pm 0.03$ and $0.77 \pm 0.01$ g/g on the grape stalk hydrolysate with and without yeast extract supplementation, respectively. Overall, similar glucose consumption rates were observed in all conditions after 24 h of growth, and *L. rhamnsosus* consumed about 80% of the available main carbon source (glucose) on media containing grape stalk hydrolysate in about 48 h.

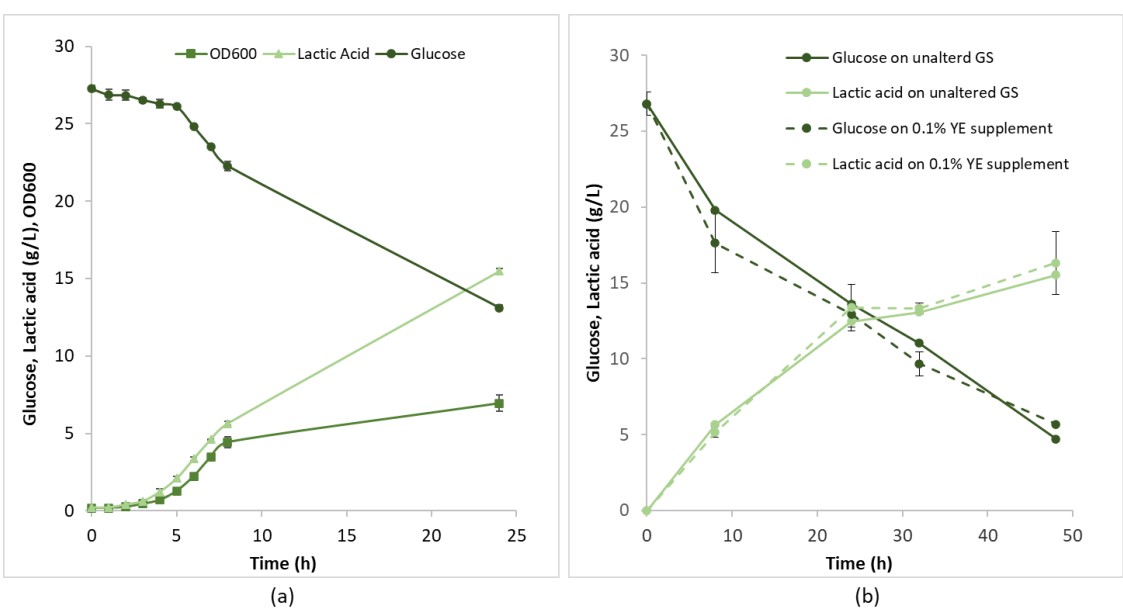

**Figure 5.** Growth of *L. rhamnosus* IMC501 on (**a**) semi-defined medium SDM and (**b**) GS hydrolysate. Concentration (g/L) of glucose and lactic acid over time.

To further evaluate the need for yeast extract supplementation and improve results obtained in uncontrolled cultivation conditions, growth of *L. rhamnosus* IMC501 was carried out in small-scale bioreactors (working volume of 2.3 L). In these conditions, the lag phase only lasted about 2–3 h, and all the glucose initially present in the medium was consumed in less than 24 h (Figure 6). Additionally, the final concentration of lactic acid produced was higher (20 g/L on average), resulting in a significant improvement in the yield that was equal to 0.98 and 0.99 g/g, respectively, in the absence and presence of yeast extract in the medium. Data confirmed a slight improvement of the glucose consumption rate in the supplemented medium (1.2 g/L·h with YE and 0.99 g/L·h without YE) and consequently of the lactic acid volumetric productivity in the first eight hours of the batch process, although the final titer of LA after 24 h of growth was highly similar. As observed in bottle experiments, xylose was not consumed on any of the media, even after glucose depletion.

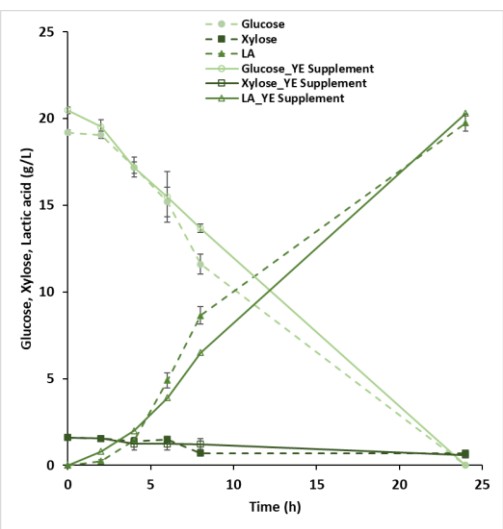

**Figure 6.** Batch fermentation of *L. rhamnosus* IMC501 on steam-exploded grape stalk hydrolysate with and without yeast extract supplementation (0.1% *w/v*). Concentration (g/L) of glucose, xylose, and lactic acid in 24 h.

The concentration of three potentially inhibitory compounds, namely furfural, vanillin, and 4-hydroxybenzoic acid, was evaluated in the steam-exploded and hydrolyzed grape stalk slurry used for small-scale bottle and fermenter experiments (Table 2). Similar concentrations of inhibitors were also found in the media at the end of the experiments after 24 or 48 h of growth.

**Table 2.** Concentration of potential inhibitors released during biomass pre-treatment in the grape stalk hydrolysates before and after strain inoculation and growth, in bottle and 3 L bioreactor experiments.

| Inhibitor | Steam-Exploded GS before Strain Inoculation (mg/L) | Steam-Exploded GS after 48 h of Growth (mg/L) |
|---|---|---|
| 4-Hydroxybenzoic acid | 12.6 ± 1.3 | 11.5 ± 0.1 |
| Furfural | 16.1 ± 0.5 | 17.7 ± 0.6 |
| Vanillin | 6.6 ± 0.25 | 6.7 ± 0.1 |

## 4. Discussion

The utilization of inexpensive and renewable biomass as a source for the production of added-value molecules, such as LA, is a major strategy to replace the petroleum-based market. In fact, raw materials can cost up to 40% of the overall LA production cost, thereby significantly affecting process sustainability [28]. Several renewable resources have been investigated to date, and those that offer an alternative to food-based feedstocks are of particular interest. Accordingly, a "zero-kilometer" economic policy directly provides waste material from local food companies, reducing pollution and promoting territorial development.

In the present work, grape stalks recovered from a wine distillery in the Campania region were evaluated as a potential source of fermentable sugars. The recalcitrance of lignocellulosic biomasses often causes poor yields of fermentable sugars and low efficiency of fermentation processes (although they are extremely convenient from an economic point of view). However, the conversion of local waste into added-value products by exploiting the fermentation capacity of certain microorganisms and their suitability for large-scale processes avoids costly waste disposal procedures and promotes sustainable biotechnological approaches.

Different pre-treatment conditions were tested, namely steam explosion, alkaline hydrolysis, and the combination of both, and results were evaluated by hydrolyzing the

pre-treated biomass with a commercial enzyme cocktail. Slightly but not significantly (two-tailed, non-homoscedastic t-student test $p > 0.05$) higher concentrations of glucose were obtained by pre-treating the biomass with ammonium hydroxide in comparison to steam explosion, whereas the sequential use of the two techniques (first steam explosion and then ammonium hydroxide) did not improve results. Alkali reagents structurally modify lignin (by degrading glycosides and side chains of esters), cause swelling and decrystallization of cellulose, and cause solvation of hemicellulose [29–32], although the effectiveness of this process largely depends on the lignin content of the biomass [33]. Compared to other chemical methods, alkaline processes use less harsh conditions, and promising results on different biomass types have been obtained so far; however, long reaction times and the neutralization of the pre-treated biomass are required [34]. On the other hand, steam explosion is one of the most efficient lignocellulosic biomass pre-treatment methods for increasing cellulose enzymatic digestibility [35]. The biomass is heated with water or steam in the temperature range of 170–230 °C for several minutes under a pressure of 8–30 bars and then subjected to a sudden pressure drop. This leads to vapor expansion inside the biomass, causing disruption of the biomass and thereby making cellulose more accessible. The structure of grape stalks treated with these two methods appears less compact compared to the untreated sample. The vascular system (xylem) is completely destructured in steam-exploded biomasses, as expected, and a complete collapse of the fibers can be observed. Exposure to ammonium hydroxide, on the other hand, caused a reduction in fiber length and a distortion of the shape of inner structural elements. Despite the different morphologies, similar results in terms of the release of fermentable sugars were achieved by steam explosion and alkaline pre-treatment of grape stalks. However, the steam-exploded biomass was used as a starting material in a series of experiments that aimed to optimize the cellulose hydrolysis protocol. In fact, the main advantages of SE are low investment costs, moderate energy consumption, and low environmental impact due to the non-utilization of acids, bases, or solvents [36]. In addition, this approach reduces the formation of microorganism-inhibiting compounds [37] and minimizes the need for chemical neutralization [38].

Different concentrations of enzymes and biomass were tested to determine the best hydrolysis operating conditions. Overall cellulose conversion efficiency ranged from about 19 to 37%. Lower enzyme: cellulose ratios corresponded to lower cellulose conversion yields. Glucose release improved significantly when 5.5% $w/w$ ($g_{enz}/g_{cellullose}$) were supplemented at time zero of the experiment; in this case, the concentration of glucose released after 48 h of incubation was 66% higher as compared to that observed after the first 24 h. Lower percentages of enzyme with respect to cellulose (2.7 (t1) and 3.6 (t5)) only resulted in a slight and not significant titer increase at the end of the experiments (48 h). This might be due to the lower ratio of available enzyme to cellulose compared to experiments performed with the same amount of enzyme mix and a lower solid loading (10% $w/v$). In all conditions tested, a two-step saccharification protocol increased glucose concentrations, indicating the presence of residual cellulose to be converted.

Atatoprak and colleagues [9] hydrolyzed sulfuric acid-pre-treated grape stalks, obtaining about 6.06 g/L of glucose after 24 h [9], with a conversion efficiency of cellulose to glucose of about 26%.

In a recent study, Spigno and co-workers [39] employed two different chemical methods for cellulose extraction from grape stalks. In particular, the acid–alkaline/oxidative (AAO) and alkaline/acid (AA) methods were compared. Here, the yield of cellulose resulted in 21.2% for AAO and 23.9% for AA, which were lower compared to those obtained in the present work. Moreover, both methods contain acid and alkaline steps that are non-environmentally friendly and not easily scalable. Although acid treatment is the most used pre-treatment method for lignocellulosic feedstocks, the high acidity causes the formation of a high amount of inhibitory chemicals, including furfurals, 5-hydroxymethylfurfural, phenolic acids, and aldehydes, that affect microbial growth. Moreover, since most acids are corrosive and dangerous, a suitable material for building the reactor that can withstand the

needed experimental conditions and the corrosive nature of acids is required. In another paper, Ping et al. [40] optimized different biomass pre-treatments with acid, organosolvent, and alkaline/oxidation. Results of the study showed that alkaline/oxidation at 170 °C for 60 min was the most effective in terms of cellulose yield, with a conversion of 45%. In this work, by extending the incubation time and identifying the optimal amount of enzyme necessary over time, up to 12 g/L of glucose from steam-exploded grape stalks were obtained with a conversion efficiency of 37% in a two-step hydrolysis protocol.

Grape stalk hydrolysate was next tested as a substrate for the growth of *L. rhamnosus* IMC501 in a series of small-scale experiments to produce lactic acid. LA is a key platform chemical, and according to the global LA market, annual production is expected to reach 1960.1 kilotons in 2025 [13]. It has applications in the food and agricultural industries, as well as in the manufacturing of pharmaceuticals, cosmetics, and chemicals. Among the most important applications of LA is the synthesis of PLA, a commonly used biodegradable and biocompatible plastic. Biotechnological production of LA allows for optically pure LA, although numerous microbial strains produce racemic mixtures of L and D-LA. Optical purity is crucial to obtain polymers with the desired mechanical properties suitable for commercial use and, at the same time, to simplify downstream purification processes that can account for 30–40% of the total production cost [41].

Growth of *L. rhamnosus* on grape stalk hydrolysate with and without supplementation of yeast extract as an additional nitrogen source was compared to that on a semi-defined medium used as a positive control. Small-scale tests highlighted the ability of the probiotic strain to grow on the renewable substrate and to convert, in the presence of yeast extract, up to 85% of the glucose into lactic acid. In all conditions, however, residual glucose was found even after 48 h of growth, probably due to the low pH caused by lactic acid accumulation.

To verify whether the additional nitrogen source could be eliminated from the cultivation medium, thereby further reducing costs, batch experiments on three-liter bioreactors were next performed. In controlled environmental conditions, the lag phase only lasted about 2–3 h; the strain did not need an adaptation phase, which is very often required in the presence of lignocellulose-derived media, indicating its ability to grow in the presence of toxic phenolic compounds released during biomass pre-treatment and hydrolysis (e.g., furfural, hydroxymethilfurfural, vanillin, etc.). The concentration of inhibitors found in the hydrolysate before strain inoculation and at the end of the growth experiments did not vary, indicating that it was not metabolized by *L. rhamnosus* IMC501. Jang and collaborators also found that inhibitors (furfural, 5-hydroxymethilfurfural, and phenol) released in seaweeds pre-treated with sulfuric acid did not affect the sugar consumption rate of *L. rhamnosus*, instead causing a reduction in the lactic acid yield [42]. This difference might be due to the higher concentration of inhibitory compounds reported by Jang et al. [42], probably because of the harsher conditions of biomass pre-treatment (acid hydrolysis) as compared to the one presented in this work (steam explosion).

About 40 and 35% of the glucose was consumed after the first eight hours of growth on the media with and without yeast extract supplementation, respectively, and no residue was found after 24 h. As for small-scale bottle experiments, xylose was not consumed. This is not surprising since the genome sequence analysis of 40 *L. rhamnosus* strains revealed that genes coding for xylose importers are part of the variome (variable genome content) and, as a result of dynamic evolution, were probably lost [33,43].

Data obtained from bioreactor experiments showed a final $Y_{LA/S}$ of $0.98 \pm 0.05$ g/g on media containing grape stalk hydrolysate only. The obtained yield of lactic acid produced on consumed glucose was high and comparable with the current scientific literature. Recently, Pontes and co-workers [44] demonstrated the production of lactic acid by growing *L. rhamnosus* ATCC 7469 on an autohydrolyzed mixture of lignocellulosic biomass (forest ecosystems) in a simultaneous saccharification and fermentation (SSF) process with a yield of 0.97 g/g. Another recent work by Radosavljevic [45] demonstrated the ability of this strain to grow on Brewer's spent grain hydrolysate with an overall yield of 0.93 g/g. Grape stalks were also used in recent studies for the production of diverse products such as

succinic acid by *Actinobacillus succinogenes* with an overall succinic acid yield of 0.67 g/g [46] and ethanol by *Pichia stipsis* with a fermentation efficiency of about 42% [9].

The results obtained in this work are quite promising since the maximal theoretical yield of lactic acid was reached. By further improving the extraction of cellulose with optimized steam explosion pre-treatments, a higher concentration of available sugars can be obtained to boost lactic acid production to industrially applicable levels.

**Author Contributions:** Conceptualization and supervision, D.C. and C.S.; writing—original draft preparation,. D.C. and S.D.; Fermentation experiments, S.D. and M.D. (Marcella D'Albore); Biomass hydrolysis, S.D., M.D. and S.S.; SE pre-treatment, L.Z. and D.B.; Cellulose and Hemicellulose determination, G.B.; SEM, M.D. (Maria D'Agostino); project administration, funding acquisition, C.S. All authors have read and agreed to the published version of the manuscript.

**Funding:** This research was funded by MUR, grant INCube.

**Institutional Review Board Statement:** Not applicable.

**Informed Consent Statement:** Not applicable.

**Data Availability Statement:** Data is contained within the article.

**Conflicts of Interest:** The authors declare no conflict of interest.

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
