# Peer review of "Grape Stalk Valorization: An Efficient Re-Use of Lignocellulosic Biomass through Hydrolysis and Fermentation to Produce Lactic Acid from Lactobacillus rhamnosus IMC501"

_fermentation, doi:10.3390/fermentation9070616_

Round 1
Reviewer 1 Report
No comments to be addressed specifically
Author Response
We kindly thank the reviewer for accepting the manuscript in its current form
Reviewer 2 Report
Overall, the introduction outlining the rationale for the study is good. However, the authors should address the following before the manuscript can be published:
Line 16: “a” should be added before building block
Line 20: 2-step hydrolysis conditions should be specified to state that enzymatic hydrolysis using cellulase was involved.
Line 23. “inhibitors” means xylose only? Since the text only measured xylose. If yes, please change to just xylose. Otherwise it is misleading.
Line 26: Keywords should not repeat those that are already in the title
Line 55. 262.3.106hL. Please put as 262.3 × 106 hL, and apply the same to the rest.
Line 109: Please add a C after 70 °
Lines 129-135. A brief explanation on why different experiments were conducted will be very helpful in allowing readers to understand and follow the methods.
Line 147. Please specify the medium.
Line 150. Please define SGLS.
Line 158. Please define % inoculum or starting OD.
Line 165. Please list the % Dissolved oxygen
Line 193. The word digestions, can be specified to indicate that the quantities of cellulose and hemicellulose stated were measured as described in Section 2.2
Lines 193-197. There seems to be a difference in structure also between steam and ammonium hydroxide treatments. The explanation in lines 315-327 should be tied to what was shown in Fig 1.
Line 249. Shouldn’t a paired t test be used instead of a student t test, when comparing means between 24 and 48 h?
Line 254. Please cite and provide evidence that it is indeed a probiotic strain with clinically evidenced health benefits.
Lines 257-258. Since there are solid residues interfering with OD measurement, were there attempts to plate instead of using OD to estimate growth?
Figures 5 and 6. Only slight improvement in lactic acid yield and glucose consumption with 0.1% YE. Could it be there was insufficient YE present? Did the authors try increasing the dosage of YE to see if there was a corresponding increase in lactic acid yield?
Line 348. Space should be added between 24 and h. This mistake is present throughout the manuscript.
Line 382. Growth should be grow.
Lines 391-392. The phenolic compounds stated were not measured. How do the authors know that these phenolic compounds were present in toxic levels, especially since steam explosion was used which produces low levels of phenolic compounds?
Lines 408-409. Please spell out the microorganisms names in full at first mention.
Minor errors with regards to spacing, and units need to be checked again.
Author Response
Line 16: “a” should be added before building block
The text was modified
Line 20: 2-step hydrolysis conditions should be specified to state that enzymatic hydrolysis using cellulase was involved.
The text was modified to: Although grape stalks are quite recalcitrant, the combination of a steam explosion pre-treatment with optimized two-step hydrolysis with commercial enzymes (Cellic-cTec2)…etc
Line 23. “inhibitors” means xylose only? Since the text only measured xylose. If yes, please change to just xylose. Otherwise it is misleading.
No, inhibitors refer to molecules that are released during the pretreatment such as phenolic compounds (e.g. vanillin, furfural etc) that can delay or even inhibit the growth of microbial strains depending on their resistance and on the concentration of compounds released in the medium. We now quantified three commonly found potential inhibitors, namely vanillin, furfural and 4-hydroxybenzoic acid, and this information was added to the manuscript in Table 2.
Line 26: Keywords should not repeat those that are already in the title
Actually the keywords that we used are a mixture of some words found in the title and some present in the text, we do not feel like removing those that are already present in the title since they are the essence of the manuscript…however new ones were now added.
Line 55. 262.3.106hL. Please put as 262.3 × 106 hL, and apply the same to the rest.
The symbol was changed as indicated
Line 109: Please add a C after 70 °
The C is already there..could there be a problem in the pdf you received?
Lines 129-135. A brief explanation on why different experiments were conducted will be very helpful in allowing readers to understand and follow the methods.
The text was changed to: In a first set of tests 24h hydrolysis was carried out on 10% (w/v) pre-treated biomasses suspended in 20 mL of sodium acetate buffer (5 mM) at pH 5.2 with 2.7 % (w/w) genz/gcellulose. These experimental conditions were used on grape stalks treated by (i) SE, (ii) alkaline hydrolysis and (iii) by the combination of SE and alkaline hydrolysis to select the most convenient approach. Successive hydrolysis tests were performed on grape stalks only pre-treated by steam explosion. In particular, experiments aimed to identify the optimal amount of biomass and of enzyme in respect to cellulose content and timing of enzyme addition (one pulse, two pulses).
Line 147. Please specify the medium.
The medium was specified.
Line 150. Please define SGLS.
Actually it’s a semidefined medium that we erroneously called SGLS; we removed SGLS and just called it SDM. The composition is reported in section 2.1.
Line 158. Please define % inoculum or starting OD.
The entire preculture was added to the fermenter, this information was added to the text.
Line 165. Please list the % Dissolved oxygen
The text was modified to: The stirring velocity was 150 rpm, and an air supply of 0.44 vvm was applied during the fermentation (Percentage of dissolved oxygen recorded by the electrode during the process: from 100 to 0 % for the first three h, 0% for the remaining 21 h of growth).
Line 193. The word digestions, can be specified to indicate that the quantities of cellulose and hemicellulose stated were measured as described in Section 2.2
The text was changed to “Triple-step digestions of grape stalks (paragraph 2.2) indicated….”
I hope I understood the reviewer’s comment.
Lines 193-197. There seems to be a difference in structure also between steam and ammonium hydroxide treatments. The explanation in lines 315-327 should be tied to what was shown in Fig 1.
This text was added in the discussion section:
The structure of grape stalks treated with these two methods appears less compact compared to the untreated sample. The vascular system (xylem) is completely destructured in steam exploded biomasses, as expected, and a complete collapse of the fibers can be observed. Exposure to ammonium hydroxide, on the other hand, caused a reduction of fibers’ length and a distortion of the shape of inner structural elements. Despite the different morphologies similar results in terms of release of fermentable sugars were achieved by steam explosion and alkaline pretreatment of grape stalks. However, the steam exploded biomass was used as starting material in a series of experiments that aimed to optimize the cellulose hydrolysis protocol. In fact, the main advantages of SE are…
Line 249. Shouldn’t a paired t test be used instead of a student t test, when comparing means between 24 and 48 h?
I normally prefer using the most restrictive conditions to be sure of data significance, but you are right, in this case it makes perfect sense to use a paired t-test. The figure was replaced.
Line 254. Please cite and provide evidence that it is indeed a probiotic strain with clinically evidenced health benefits.
References for the strain that refer to the commercial product (SYNBIO) containing L. rhamnosus IMC501 were added in the introduction.
Lines 257-258. Since there are solid residues interfering with OD measurement, were there attempts to plate instead of using OD to estimate growth?
Growth as such in terms of biomass production was not estimated at all, it was measured only during growth on the control semidefined medium (SDM). Based on previous experience with agro-industrial waste we preferred to only consider glucose consumption and lactic acid production as parameter to evaluate indirectly strain growth. Viability counts on plates are also resulting in quite unreliable results due to the inhomogeneity of the broth.
Figures 5 and 6. Only slight improvement in lactic acid yield and glucose consumption with 0.1% YE. Could it be there was insufficient YE present? Did the authors try increasing the dosage of YE to see if there was a corresponding increase in lactic acid yield?
We did not consider increasing the concentration of yeast extract for two reasons: the first one is because the idea was to simplify the medium ideally using only hydrolysates without the addition of any other component thereby reducing costs and simplifying preparation procedures; the second one is that actually the yield of product on substrate is already so close to the maximal theoretical yield that we are not expecting a huge improvement in controlled bioreactor conditions.
Line 348. Space should be added between 24 and h. This mistake is present throughout the manuscript.
The manuscript was revised.
Line 382. Growth should be grow.
No we meant to use the noun not the verb.
Lines 391-392. The phenolic compounds stated were not measured. How do the authors know that these phenolic compounds were present in toxic levels, especially since steam explosion was used which produces low levels of phenolic compounds?
We now measured the concentration of 3 compounds and added a table (table 2).We also added comments in the discussion section:
In controlled environmental conditions the lag phase only lasted about 2-3 h, the strain did not need an adaptation phase that is very often required in the presence of lignocellulose derived media, indicating its ability to grow in the presence of toxic phenolic compounds released during biomass pre-treatment and hydrolysis (e.g. furfural, hydroxymethilfurfural, vanillin etc.). The concentration of inhibitors found in the hydrolysate before strain inoculation, and at the end of the growth experiments, did not vary, indicating that it was not metabolized by L. rhamnosus IMC501. Jang and collaborators found that also inhibitors (furfural, 5-hydroxymethilfurfural and phenol) released in sulfuric acid pre-treated seaweeds did not affect the sugar consumption rate of L. rhamnosus, causing on the other hand a reduction of the lactic acid yield [42]. This difference might be due to the higher concentration of inhibitory compounds reported by Jang et al [42] probably due to harsher conditions of biomass pre-treatment (acid hydrolysis) as compared to the ones presented in this work (steam explosion).
Lines 408-409. Please spell out the microorganisms names in full at first mention.
The text was revised

Reviewer 3 Report
1. In the Abstract, the authors mentioned that “L. rhamnosus IMC501 can tolerate biomass-derived inhibitors and grow on grape stalk hydrolysate” (lines 22-23), but they did not present the results of the analysis in the section Results, neither they comment on the inhibitors concentration in Discussion.”
2. References are not written according to the instructions to the authors. Please write the references according to journal guidelines for authors.
3. The manuscript suffers from errors such as typos, sentence structure, and non-scientific words that should be corrected and reworded.
4. Pretreatment conditions (temperature and pressure) should be described in Materials and Methods. Figure 1 should be moved to subsection 2.3.
5. Extremely high concentrations of enzymes were used in experiments. According to the authors, cellulase dosage was between 2.7 and 5.5 g of enzyme preparation/g of cellulose. The activity of the Cellic CTech2 is 127.5 ± 4.6 FPU/g https://dspace.lib.cranfield.ac.uk/bitstream/handle/1826/15255/Augmented_hydrolysis_of_acid_pretreated_sugarcane_bagasse-2020.pdf?sequence=4#:~:text=Enzyme%20and%20its%20protein%20determination&text=CTec2%20was%20found%20to%20be,127.5%20%C2%B1%204.6%20FPU%2Fg or 110 FPU per ml according to literature (see: https://pubs.rsc.org/en/content/articlepdf/2017/ra/c7ra02477k). This means that enzyme loading in this research was between 297 and 605 FPU/g of cellulose, making the processes expensive and economically unfeasible. Typical cellulase loading in experiments is usually between 10 and 30 FPU/g of cellulose.
6. Despite the high enzyme loading, sugar yield is rather low, which raises the question regarding the correctness of analytical methods and calculations. Please check the analytical methods and all calculations.
7. Data on the composition of pretreated biomass are missing. Please analyze cellulose and hemicellulose content and add the results to the manuscript.
8. The data on glucose concentrations after 48 h of enzyme hydrolysis of pretreated LB are presented twice in Figure 4 and Table 1. All data should be presented only once in a figure or table.
9. Lines 264-271:
Since the initial concentration of the carbon source (glucose) in these experiments was different, the product yield in these cultivations should not be compared. Therefore, the reviewer suggests deleting this text. The reviewer suggests conducting the control experiment using the same cultivation medium in which LB hydrolysate (carbon source) would be replaced with pure glucose. The concentration of glucose should be equal to that in LB hydrolysate.
Author Response
- In the Abstract, the authors mentioned that “L. rhamnosus IMC501 can tolerate biomass-derived inhibitors and grow on grape stalk hydrolysate” (lines 22-23), but they did not present the results of the analysis in the section Results, neither they comment on the inhibitors concentration in Discussion.”
The concentration of some of the potential inhibitors were now measured and data were added to the manuscript in table 2 and in the discussion section:
In controlled environmental conditions the lag phase only lasted about 2-3 h, the strain did not need an adaptation phase that is very often required in the presence of lignocellulose derived media, indicating its ability to grow in the presence of toxic phenolic compounds released during biomass pre-treatment and hydrolysis (e.g. furfural, hydroxymethilfurfural, vanillin etc.). The concentration of inhibitors found in the hydrolysate before strain inoculation, and at the end of the growth experiments, did not vary, indicating that it was not metabolized by L. rhamnosus IMC501. Jang and collaborators found that also inhibitors (furfural, 5-hydroxymethilfurfural and phenol) released in sulfuric acid pre-treated seaweeds did not affect the sugar consumption rate of L. rhamnosus, causing on the other hand a reduction of the lactic acid yield [42]. This difference might be due to the higher concentration of inhibitory compounds reported by Jang et al [42] probably due to harsher conditions of biomass pre-treatment (acid hydrolysis) as compared to the ones presented in this work (steam explosion).
- References are not written according to the instructions to the authors. Please write the references according to journal guidelines for authors.
Although instructions to authors only require editing of the references according to the journal style after acceptance of the manuscript, we did use the template reported on the fermentation homepage even for the initial submission:
Journal Articles:
1. Author 1, A.B.; Author 2, C.D. Title of the article. Abbreviated Journal Name Year, Volume, page range.
- The manuscript suffers from errors such as typos, sentence structure, and non-scientific words that should be corrected and reworded.
The manuscript was revised
- Pretreatment conditions (temperature and pressure) should be described in Materials and Methods. Figure 1 should be moved to subsection 2.3.
As indicated by the reviewer the figure (2) indicating temperature and pressure during steam explosion was moved to section 2.3 and pre-treatment conditions were now specified in the materials and methods section.
- Extremely high concentrations of enzymes were used in experiments. According to the authors, cellulase dosage was between 2.7 and 5.5 g of enzyme preparation/g of cellulose. The activity of the Cellic CTech2 is 127.5 ± 4.6 FPU/g https://dspace.lib.cranfield.ac.uk/bitstream/handle/1826/15255/Augmented_hydrolysis_of_acid_pretreated_sugarcane_bagasse-2020.pdf?sequence=4#:~:text=Enzyme%20and%20its%20protein%20determination&text=CTec2%20was%20found%20to%20be,127.5%20%C2%B1%204.6%20FPU%2Fg or 110 FPU per ml according to literature (see:https://pubs.rsc.org/en/content/articlepdf/2017/ra/c7ra02477k). This means that enzyme loading in this research was between 297 and 605 FPU/g of cellulose, making the processes expensive and economically unfeasible. Typical cellulase loading in experiments is usually between 10 and 30 FPU/g of cellulose.
Thank you very much for the observation, in fact we misreported the amount of enzyme used…what we indicated as 2.7, 5.5, 3.6 etc genzyme/gcellulose is a percentage % w/w (genzyme/gcellulose), meaning that 2.6 corresponds to 0.026 genzyme/gcellulose and the same for the other ones.. basically by error we indicated the amount of enzyme to be used in 100 g of cellulose!!!!
We now corrected the units from genzyme/gcellulose to % w/w (genzyme/gcellulose).
The amounts of enzyme that we used were selected to be in the mid range suggested by Cellic-Ctec 2 manufacturers as reported in the guide below:
- Despite the high enzyme loading, sugar yield is rather low, which raises the question regarding the correctness of analytical methods and calculations. Please check the analytical methods and all calculations.
Please see previous answer.
- Data on the composition of pretreated biomass are missing. Please analyze cellulose and hemicellulose content and add the results to the manuscript.
Cellulose and hemicellulose were analysed as described in the materials and methods section, paragraph 2.2. The results are reported in paragraph 3.2 first line.
- The data on glucose concentrations after 48 h of enzyme hydrolysis of pretreated LB are presented twice in Figure 4 and Table 1. All data should be presented only once in a figure or table.
Data indicating glucose released after 48 h of incubation were removed from the table, as suggested by the reviewer, to avoid repetitions.
- Lines 264-271:
Since the initial concentration of the carbon source (glucose) in these experiments was different, the product yield in these cultivations should not be compared. Therefore, the reviewer suggests deleting this text. The reviewer suggests conducting the control experiment using the same cultivation medium in which LB hydrolysate (carbon source) would be replaced with pure glucose. The concentration of glucose should be equal to that in LB hydrolysate.
As suggested by the reviewer experiments with exactly the same initial concentration of glucose in the control medium were performed in triplicate and the graph in the figure was replaced by the new one.
The yield is slightly higher, the text was modified accordingly.

Round 2
Reviewer 3 Report
The authors have improved the manuscript according to the reviewer's comments.